# Incidence of Congenital Muscular Torticollis in Babies from Southern Portugal: Types, Age of Diagnosis and Risk Factors

**DOI:** 10.3390/ijerph19159133

**Published:** 2022-07-26

**Authors:** Beatriz Minghelli, Noémia Guerreiro Duarte Vitorino

**Affiliations:** 1Escola Superior de Saúde Jean Piaget Algarve, Instituto Piaget, 8300-025 Silves, Portugal; 2Research in Education and Community Intervention (RECI), Instituto Piaget, 1950-157 Lisbon, Portugal; 3Algarve University Hospital Center (CHUA), Physical Medicine and Rehabilitation Service, 8500-338 Portimão, Portugal; noemia.vitorino@chalgarve.min-saude.pt

**Keywords:** congenital muscular torticollis, babies, incidence, epidemiology, risk factor

## Abstract

Congenital torticollis (CMT) is the most common type of torticollis and is defined as a unilateral contracture of the sternocleidomastoid muscle resulting in lateral head tilt associated with contralateral rotation, and early detection and treatment may present a high probability of recovery of head posture symmetry. This study aimed to verify the incidence of torticollis in babies born in southern Portugal types, age of diagnosis and the risk factors. This study comprised 6565 infants born in the south of Portugal at Algarve University Hospital Center, Portimão unit during a 5-year period (January 2016 to December 2020). The cases diagnosed with torticollis referred to the Pediatrics and Pediatric Physiatrist consultations at this hospital were included. 118 babies—77 (65.3%) male and 41 (34.7%) female—were diagnosed with torticollis. The incidence of a 5-year period was 1.5%. Spontaneous vaginal delivery was prevalent (*n* = 56; 47.5%), with 106 (89.8%) deliveries with cephalic presentation. 53 (44.9%) cases of torticollis were classified as postural, 37 (31.4%) as muscular torticollis with joint limitation and 28 (23.7%) as congenital torticollis (with the presence of a nodule). Postural torticollis was diagnosed at an average age of 70.14 days, muscular torticollis with joint limitation at an average of 64.12 days and congenital torticollis at 33.25 days (*p* < 0.001). Plagiocephaly was present in 48 (40.7%) babies with torticollis (*p* = 0.005) and joint limitation in 53 (44.9%) babies (*p* < 0.001). The data obtained revealed a low incidence of CMT, with the majority being classified as postural. The age of diagnosis varied between 33 to 70 days from birth. The baby’s gender, mode of delivery and the fetal presentation during delivery did not show a statistically significant association with the presence of torticollis. Despite presenting a low incidence, it is important to mention the importance of professional health intervention in the implementation of prevention strategies.

## 1. Introduction

Congenital torticollis (CMT) is one of the most common pediatric musculoskeletal conditions seen in infancy, defined as a unilateral contracture of the sternocleidomastoid muscle resulting in lateral tilt of the head associated with contralateral rotation [1,2,3,4,5,6,7,8,9,10].

The incidence of CMT varies between 0.3 and 2% [1,2,6], being more prevalent in males, with a ratio of 3:2 [1,6], and more common on the right side [1,6,7]. However, these data present discrepancies between the studies, since the investigation by Stellwagen et al. [11] found that 16% of newborns had torticollis.

Diagnosis of CMT is made via observation of alignment, cervical range of motion assessment and palpation [12]. There are 3 types of CMT: torticollis with a tumor or a palpable fibrotic nodule in the sternocleidomastoid muscle, torticollis with rigidity of the sternocleidomastoid muscle, but without an associated tumor, called muscular torticollis and postural torticollis, which consists of congenital torticollis with all the clinical features of torticollis, but without rigidity and tumor in the sternocleidomastoid muscle [13]. Children with CMT present with a cervical muscle strength imbalance [14].

There are several theories for the etiology of CMT, but it is still not fully known; it is believed to be attributed to fetal head descent or abnormal intrauterine fetal positioning during the third trimester, resulting in trauma to the sternocleidomastoid muscle. Other theories refer to fibrosis of the sternocleidomastoid muscle as a cause [8,13], resulting from venous occlusion due to persistent intrauterine lateral flexion and rotation of the neck, or trauma to the sternocleidomastoid muscle during delivery [4,13,15]. Other risk factors involved in the development of congenital torticollis include breech presentation, multiple pregnancy and dystocic delivery (vaginal delivery using suction cups or forceps) [4,6,16].

CMT may be associated with one or more comorbidities; some of these comorbidities are: brachial plexus injury, hip dysplasia, limb deformity and early developmental delay, facial asymmetry, plagiocephaly, and temporomandibular joint disorder [2,4,8]. Several studies report an association between the presence of CMT and the development of hip dysplasia with at a rate that can go up to 20% [6,17].

Early detection and treatment of CMT (before 1 month of age) has a high probability of recovering head posture symmetry (1.5 months). If CMT is detected from 6 months of age, the range of motion in the cervical region will gradually decrease, and a longer intervention period (between 9 and 10 months) may be necessary [1,8].

There are no recent studies about incidence of CMT. The aim of this study was to examine incidence of CMT at our institution, type distribution, infant’s age at a diagnosis and associated factors.

## 2. Materials and Methods

### 2.1. Design

This is a longitudinal retrospective study that comprised 6565 infants born in the south of Portugal (Algarve region) between January 2016 and December 2020 at Algarve University Hospital Center (CHUA), Portimão unit, Portugal.

The Algarve region occupies an area of 4996 km^2^ and is home to 467,495 inhabitants (2021), concentrating 4.5% of the resident population in Portugal, which comprises only one subregion, consisting of 16 municipalities and divided into 67 parishes.

Under the public health system, there are 2 university hospitals (CHUA) with delivery and maternity rooms: one located in the city of Portimão and the other in Faro, Portugal. On a private level, there is only one hospital in Faro.

### 2.2. Cases

Among all children born during the 5-year period, we selected cases diagnosed with torticollis referred to the Pediatrics and Pediatric Physiatrist consultations at the Algarve University Hospital Center, Portimão unit. The diagnosis was made by a physiatrist.

This study was approved by the Ethics Committee for health and by the Board of Directors of Algarve University Hospital Center, Portimão unit (Reference: 01/21 of 1 June 2021).

### 2.3. Data Analysis

Statistical analysis was performed using the Statistical Package for the Social Sciences (SPSS) software, version 28.0. (Armonk, NY, USA: IBM Corp).

First, data referring to the descriptive statistics of all the study variables were obtained, through measures of central tendency and dispersion. The Chi-Square Independence test, using contingency tables, was used for the statistical inference of qualitative variables (presented in Table 3). In all inferential analyses, statistical significance was set at 0.05.

Incidence Proportion (IP) was computed through the division of the total number of infants who had torticollis by the number of children born each year and in the last 5 years [18].

## 3. Results

Between January 2016 and December 2020, 6565 live births occurred at Algarve University Hospital Center, Portimão unit. Among them, 118 babies (*n* = 77; 65.3% male and *n* = 41; 34.7% female) were diagnosed with torticollis in Pediatrics and Pediatric Physiatrist consultations (incidence = 1.5%, and 95% confidence interval (CI) 1.3–1.8%).

Table 1 show the incidence proportion in each year analyzed and the cumulative incidence by a period of 5 years.

The average gestational weeks was 37.7 (SD: 2.9), with the shortest period being 28 weeks and the longest being 41 weeks.

Most deliveries of babies diagnosed with torticollis were spontaneous vaginal delivery (56; 47.5%), 43 (36.4%) cesarean section, 15 (12.7%) dystocic with the use of a suction cup and 3 (2.5%) with the use of forceps, with 1 delivery mode could not be identified in the process.

The fetal presentation was cephalic in most deliveries (106; 89.8%), with only 9 (7.6%) with breech presentation and 3 (2.5%) without reference in the process. Some deliveries were of twins (9; 7.6%), but the majority were of the birth of one baby (109; 92.4%).

Regarding the type of torticollis, 53 (44.9%) were classified as postural (congenital torticollis with all the clinical features of torticollis, but without rigidity and tumor in the sternocleidomastoid muscle), 37 (31.4%) as muscular torticollis with joint limitation (torticollis with rigidity of the sternocleidomastoid muscle, but without an associated tumor) and 28 (23.7%) as congenital torticollis with the presence of a nodule (torticollis with a tumor or a palpable fibrotic nodule in the sternocleidomastoid muscle).

The mean age of consultation at which babies were diagnosed with torticollis was 83.79 days (SD: 45.23), with a minimum age of 9 days and a maximum of 180 days.

Postural torticollis was diagnosed at an average age of 70.14 days, muscular torticollis with joint limitation at an average of 64.12 days and congenital torticollis at 33.25 days (*p* < 0.001).

As for the laterality of torticollis, 57 (48.3%) were diagnosed on the right and 61 (51.7%) on the left. The dominant gender in CMT is male, with 77 (65.3%) and 41 (34.7%) females.

Plagiocephaly was present in 48 (40.7%) of the babies with torticollis and absent in 63 (53.4%), with 7 (5.9%) without this reference in the process. As for the laterality of plagiocephaly, 27 (56.3%) were on the right, 19 (39.6%) on the left and 2 (4.1%) were symmetrical.

Table 2 presents the association between plagiocephaly and laterality of the torticollis.

Regarding the presence of limited range of motion, it was present in 53 (44.9%) babies diagnosed with torticollis, absent in 60 (50.8%) babies and without reference in 5 (4.2%), 28 (52.8%) on the right side and 25 (47.2%) on the left side.

Table 3 shows the association between the different types of torticollis and the qualitative variables analyzed in the study.

## 4. Discussion

The data presented revealed a low incidence of CMT in babies who went to the consultation in the period of 5 years (1.5%), with 2018 being the year with the highest incidence with 2.7% and in 2017 there was a lower incidence with 1.3%. Similar data were found in the study by Cheng et al. [14], who verified 624 cases of torticollis in babies born in China over a period of 7 years and whose incidence was 1.3%.

Considering only the babies who were diagnosed with torticollis, most were born by spontaneous vaginal delivery (48%), but it was not possible to fulfill the applicability conditions of the statistical test to verify the possibility of obtaining statistical significance. These data are contrary to those found in studies carried out in the center (Coimbra), between 2008 and 2011 [19], and in the north of Portugal (Porto), between 2012 and 2014 [4], where it was found that dystocic delivery was predominant with values of 73%, in both studies, with a higher incidence of delivery instrumented by cupping or forceps.

The sternocleidomastoid stretching muscle during childbirth may be a direct cause of CMT; however, no studies were found that showed the relationship between mode of childbirth and torticollis. Hardgrib et al. [20] and Lee et al. [21] compared vaginal deliveries with cesarean sections and found no difference in the clinical severity of CMT according to the mode of delivery, suggesting that prenatal factors can probably cause CMT due to the reduced risk of birth trauma in cesarean sections. Ho et al. [22] found higher rates of assisted breech deliveries, instrumental deliveries and cesarean sections, which led them to conclude that birth trauma could be the main etiological factor of CMT.

The cephalic presentation was verified in most deliveries (90%) and only 8% of babies diagnosed with torticollis were born with breech presentation, lower data than those presented in national studies in which breech presentation was verified in 21% of babies with torticollis [4,19]. The low number of births with breech presentation may be due to the fact that there is a growing tendency among obstetricians to recommend cesarean delivery when the fetus is a breech presentation [19].

Most babies diagnosed with torticollis were male (65%), data similar to those found in the study by Bastos et al. [19] (67%) and in the study by Petronic et al. [1]. However, in the study by Amaral et al. [4], there was no big difference in gender, being 51% male and 49% female. Despite the difference between the percentages of the gender observed in this study, there was no statistically significant association with the presence of torticollis.

Regarding the classification of torticollis, postural torticollis was the most prevalent (45%), data that differ from the results obtained in other studies, and which present a greater number of babies classified with congenital torticollis and with the presence of a nodule and muscular torticollis [19,23]. Cheng et al. [23] found a higher percentage of congenital torticollis with tumor (55%), compared to muscular (34%) and postural (11%) torticollis in 821 babies, and the data obtained in the study by Ho et al. [22] revealed a 36% prevalence of torticollis with a tumor in 91 babies. The study by Amaral et al. [4] verified the presence of torticollis with the presence of a nodule in 13% of the babies, and muscular torticollis was observed in 21%. The reliability of the diagnosis of the type of torticollis may play a major role in the differences between studies. For example, the clinician’s ability for observation the presence of neck and/or facial or cranial asymmetry, to palpate using passive cervical rotation the SCM, can also be crucial to a good diagnosis [10].

Postural torticollis seems to be associated with a preference for head posture at birth, due to the presence of a deformational plagiocephaly, and aggravated by placing the baby most often in the same position, in the first months, when they have poor head control [24].

As for laterality, a difference of only 3.4% was observed between the right and left sides, the latter being the most prevalent (51.7%), data very close to those obtained in the study by Bastos et al. [19] (52.8% of the cases were on the left side) and those of Ho et al. [22] (50.5%). The study by Amaral et al. [4] observed an equal distribution between the sides. The study by Petronic et al. [1] revealed a greater right laterality predominance, but also with little difference in percentages terms.

Plagiocephaly was present in less than half of babies with torticollis (41%), with a greater predominance in muscular torticollis (64%). Data obtained in central Portugal region [19] showed a higher percentage of plagiocephaly associated with torticollis (51%). Regarding the laterality of plagiocephaly obtained, in this study, the right side is the most predominant (57%), confirming the prevalence of torticollis on the left (47.5%). Peitsch et al. [25] evaluated a sample of 201 newborns and found that 13.1% of the babies had lateral or posterior cranial flattening, with 54% having flattening on the right side, with a predominance of males, suggesting that boys have larger heads and less flexible than girls, which makes them more susceptible to head deformity anomalies at birth [26]. There is still the perception that torticollis develops secondary to deformational plagiocephaly, due to the maintenance of the head position, in a “comfort position” [25].

The mean age at the time of diagnosis of babies with torticollis was less than 3 months of age. Data from the study Ho et al. [22], which evaluated the presence of torticollis that occurred between 1994 and 1997, revealed that the mean diagnosis was 2 months of age. Bastos et al. [19] revealed that the first diagnostic consultation took place between 2 weeks of age and until almost 4 years of age, most of them being performed with babies aged less than 6 months (77%) and in the study by Amaral et al. [20], the mean was lower at 3 months (11.6 weeks).

Most congenital cases of torticollis were diagnosed in babies aged up to 1 month (57%), and most postural torticollis cases were diagnosed in babies aged over three months (45%), with muscular torticollis diagnosed at ages above 2 months (65%). According to the Academy of Pediatric Physical Therapy’s guide, infants with CMT should be referred to a physiotherapist as soon as they are identified [10].

Parental education to prevent asymmetries/CTM, the assessment and early identification of CMT (the identification of infants who have postural preference, reduced cervical range of motion, sternocleidomastoid masses, and/or craniofacial asymmetry) are prevention strategies. Prevention consists of an anticipated action, based on epidemiological knowledge to control and reduce the risk of diseases. In this way, prevention and education projects are based on scientific information and normative recommendations, in addition, investments in prevention are always less expensive than those applied in the management and treatment of the disease [27].

This study presents some limitations. For bureaucratic reasons, there may have been some delays in scheduling appointments, not allowing, in some cases, an earlier diagnosis, which may compromise the data on the age of the baby’s diagnosis. Furthermore, some infants were not referred for consultation in the Algarve University Hospital Center, Portimão unit and have been diagnosed in another hospital or clinic, thus not being counted in the study. Another limitation is the fact that some babies may have been born in this hospital, but not have gone to the pediatric consultation in this hospital. Examples: at the time of delivery this was the closest hospital, but the parents did not live here; some parents may have changed residence and gone to another location for the pediatric consultation; and some parents may have preferred to go to the consultation at a clinic or hospital particular.

## 5. Conclusions

The data obtained in this study verified an incidence of congenital torticollis of 1.5% in 6565 babies born in a hospital in the south of Portugal between 2016 and 2020, the most prevalent being postural CMT. The baby’s gender, mode of delivery and the fetal presentation during delivery did not show a statistically significant association with the presence of torticollis.

Since most postural torticollis have been diagnosed in babies over the age of three months, there is an interval of opportunity for intervening as early as possible to prevent this type of torticollis.

## Figures and Tables

**Table 1 ijerph-19-09133-t001:** Incidence proportion by years and cumulative incidence.

Year of data collection	2016	2017	2018	2019	2020	5-years period
**Numbers of births**	1283	1302	1359	1363	1258	6565
**Diagnosed cases**	25	17	22	31	23	118
Incidence proportion	0.019	0.013	0.027	0.023	0.018	0.015
	CI: 0.013–0.029	CI: 0.008–0.021	CI: 0.019–0.037	CI: 0.016–0.032	CI: 0.012–0.027	CI: 0.013–0.018

**Table 2 ijerph-19-09133-t002:** Relationship between the presence of plagiocephaly and the laterality of the torticollis.

Plagiocephaly	Laterality of Torticollis		Total
	Right	Left	
**Absent**	34 (59.6%)	29 (47.5%)	63 (53.4%)
**Present**	19 (33.3%)	29 (47.5%)	48 (40.7%)
**Without reference**	4 (7%)	3 (4.9%)	7 (5.9%)

**Table 3 ijerph-19-09133-t003:** The association between the torticollis types and the qualitative variables.

Variables		Types of Torticollis			Total	*p*-Value
		Congenital	Muscular with Joint Limitation	Postural		
**Gender**	Male	18 (64.3%)	26 (70.3%)	33 (62.3%)	77 (65.3%)	0.729
	Female	10 (35.7%)	11 (29.7%)	20 (37.7%)	41 (34.7%)	
**Delivery mode**	Spontaneous vaginal delivery	11 (39.3%)	18 (50%)	27 (50.9%)	56 (47.9%)	*
	Caesarean	10 (35.7%)	14 (38.9%)	19 (35.8%)	43 (36.8%)	
	Dystocic	7 (25%)	4 (11.1%)	7 (13.2%)	18 (15.4%)	
**Fetal presentation**	Cephalic	26 (96.3%)	39 (82.9%)	51 (96.2%)	106 (92.2%)	*
	Pelvic	1 (3.7%)	6 (17.1%)	2 (3.8%)	9 (7.8%)	
**Age at time of diagnosis**	Until 1 month	16 (57.1%)	5 (13.5%)	5 (9.4%)	26 (22%)	<0.001
	0 up to 2 months	6 (21.4%)	8 (21.6%)	11 (20.8%)	25 (21.2%)	
	2 up to 3 months	2 (7.1%)	12 (32.4%)	13 (24.5%)	27 (22.9%)	
	3 months and older	4 (14.3%)	12 (32.4%)	24 (45.3%)	40 (33.9%)	
**Laterality**	Right side	13 (46.4%)	19 (51.4%)	25 (47.2%)	57 (48.3%)	0.903
	Left side	15 (53.6%)	18 (48.6%)	28 (52.8%)	61 (51.7%)	
**Plagiocephaly**	Absent	19 (76%)	13 (36.1%)	31 (62%)	63 (56.8%)	0.005
	Present	6 (24%)	23 (63.9%)	19 (38%)	48 (43.2%)	
**Limited range of motion**	Absent	10 (38.5%)	0	50 (100%)	60 (53.1%)	<0.001
	Present	16 (61.5%)	37 (100%)	0	53 (46.9%)	

* Does not meet the applicability conditions of the chi-square independence test.

## Data Availability

The data are restricted to Algarve University Hospital Center, Portimão unit and cannot be provided to others.

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
