# Peer review of "Incidence of Congenital Muscular Torticollis in Babies from Southern Portugal: Types, Age of Diagnosis and Risk Factors"

_ijerph, 2022, doi:10.3390/ijerph19159133_

Round 1

Reviewer 1 Report

The topic of torticollis in infants is always interesting to pediatric physical therapists.  In addition, any changes in the trends or differences due to medical, social or cultural practices would be of interest.  However, in this manuscript, vital information is missing that may help explain the findings. 

For example, it appears that the incidence is low - similar to those before supine positioning was recommended in the US.  Are positioning practices similar in Portugal?  Also, do the healthcare practices ensure that all of the infants born in the initial cohort would be followed by the same practice throughout infancy?  Might any have gone elsewhere for diagnosis and treatment of torticollis? 

Most of the articles cited are relevant but there are many that are much more current that are not utilized to indicate more current statistics.  Many of the studies cite, although classic, are approximately 15-20 years old.  Others that are omitted have been published within the past 5 years. In the same line of questions is the notable omission of the Clinical Practice Guideline article published by the American Physical Therapy Association Pediatric Academy.  Although not a primary research article, it was a synthesis of the most current (2018) culmination of information. 

The statistical analysis description appeared to be quite thorough and the statistics presented were interesting if given the information requested above.  Of particular interest is the age at diagnosis which makes the reader question who is doing the screening, how they screen and if knowledge translation might allow improved identification and intervention?

In the introduction, the author cites Cheng as indicating that a child with congenital muscular torticollis has a muscle imbalance with greater strength on the injured side.  Did Cheng actually state that the muscle is stronger or is the muscle simply shorter?  This may lead to some confusion.

In the abstract, the author defines and uses the common abbreviation CMT and later uses CMD without a definition - is this a typo?  If not, it should be defined. 

The results and conclusion sections contain several run-on sentences an should be re-worked for grammatical errors. 

In addition, the authors assert that the postural torticollis that was seen could  be prevented by positioning practices and use of containers despite the fact that these practices were not studied.  These conclusions outreach the data.

In the last discussion paragraph, just prior to the conclusion, this sentence/phrase is confusing and contains sentence fragments, "Muscular torticollis occurs as a consequence of in utero restriction or prolonged attachment of the head of the head. fetus, resulting in a muscular imbalance of the cervical spine."

Author Response

Thank you for affording us another opportunity to make our article read better for the global audience. We appreciate all the comments and suggestions that were made by two reviewers. I hope we have made the revisions to your satisfaction.

Reviewer 1:

The topic of torticollis in infants is always interesting to pediatric physical therapists.  In addition, any changes in the trends or differences due to medical, social or cultural practices would be of interest.  However, in this manuscript, vital information is missing that may help explain the findings. 

For example, it appears that the incidence is low - similar to those before supine positioning was recommended in the US.  Are positioning practices similar in Portugal?  Also, do the healthcare practices ensure that all of the infants born in the initial cohort would be followed by the same practice throughout infancy?  Might any have gone elsewhere for diagnosis and treatment of torticollis? 

Answer: The recommendations here are also in supine, but we have no way of controlling if this is done, no one can control it, we are dependent on the parents' answers which may not correspond to the truth either. We added this: “The study has other limitation, as the fact that some babies were not referred for consultation in the Algarve University Hospital Center, Portimão unit, having been diagnosed in another hospital or clinic, thus not being counted in the study. Another limitation is the fact that some babies were born in this hospital because the mother was here at the time of delivery or some parents may have changed residence and gone to another place for the pediatric consultation.”

Most of the articles cited are relevant but there are many that are much more current that are not utilized to indicate more current statistics.  Many of the studies cite, although classic, are approximately 15-20 years old.  Others that are omitted have been published within the past 5 years. In the same line of questions is the notable omission of the Clinical Practice Guideline article published by the American Physical Therapy Association Pediatric Academy.  Although not a primary research article, it was a synthesis of the most current (2018) culmination of information. 

Answer: The study is about the incidence of torticollis, not about clinical practices and forms of diagnosis. There are some articles on torticollis, but they do not have data on its incidence, addressing other issues that do not meet our objective.

The statistical analysis description appeared to be quite thorough and the statistics presented were interesting if given the information requested above.  Of particular interest is the age at diagnosis which makes the reader question who is doing the screening, how they screen and if knowledge translation might allow improved identification and intervention?

Answer: We added this: “The diagnosis was made by a physiatrist.”

In the introduction, the author cites Cheng as indicating that a child with congenital muscular torticollis has a muscle imbalance with greater strength on the injured side.  Did Cheng actually state that the muscle is stronger or is the muscle simply shorter?  This may lead to some confusion.

Answer: Withdraw.

In the abstract, the author defines and uses the common abbreviation CMT and later uses CMD without a definition - is this a typo?  If not, it should be defined. 

Answer: It was a mistake. Got corrected.

The results and conclusion sections contain several run-on sentences an should be re-worked for grammatical errors. 

In addition, the authors assert that the postural torticollis that was seen could  be prevented by positioning practices and use of containers despite the fact that these practices were not studied.  These conclusions outreach the data.

Answer: This paragraph has been taken from the conclusion.

In the last discussion paragraph, just prior to the conclusion, this sentence/phrase is confusing and contains sentence fragments, "Muscular torticollis occurs as a consequence of in utero restriction or prolonged attachment of the head of the head. fetus, resulting in a muscular imbalance of the cervical spine."

Answer: Adjusted.

Reviewer 2 Report

I am excited by the aims of this article and all the work that was put into it. I believe that the topic is relevant and important to the field of pediatrics. However, the introduction, results and discussion are not well structured and need a lot of work to make it read and look professional. I recognize that English is not authors' first language, but the flow of the article itself needs major improvements.

Introduction: It reads a bit awkward and choppy. 

First sentence: you say that it is the most common type of torticollis (what are the other types?). It may be better to state that It is one of the most common pediatric musculoskeletal conditions observed in infancy. You later discuss different types of CMT which may confuse the reader with the types mentioned in the first sentence.

Combine second and third sentence and improve the flow. It is a main point of your research so it would make it stronger to focus on the incidence first and maybe later talk about boys to girls ratio. Was there is difference in how Stellwagen defined CMT vs other authors. Why is his incidence so high vs others? You can use it in your discussion as well.

4th sentence (line 38). Consider: Diagnosis of CMT is made via observation of alignment, cervical range of motion assessment, and palpation.

39. This is a very confusing sentence: consider making it more streamlined and clear: For example: There are 3 types of CMT: postural, muscular and muscular with SCM mass.

Line 40. Consider changing word "rigidity" for "decreased flexibility)

Line 44. needs grammar edits. Consider: Is is greater strength or is the muscle overactive? Or both? This paragraph is only one sentence. Paragraph needs more than 2 sentences to be a paragraph. Consider combing with the section above.

Line 46. there are several theories not just one. Your first sentence is misleading.  Rework this paragraph for better flow. Consider children who acquire CMT after birth, not in prenatal period. Your results show infants diagnosed later right?

Line 54-57. Needs to be edited for grammar and structure. Consider adding to your citations article by von Heideken at al.

von Heideken J, Green DW, Burke SW, Sindle K, Denneen J, Haglund-Akerlind Y, Widmann RF. The relationship between developmental dysplasia of the hip and congenital muscular torticollis. J Pediatr Orthop. 2006 Nov-Dec;26(6)

Lines 59-61. rewrite for grammar and clarity. It is confusing, especially 2nd sentence. Consider: is it about detection or initiation of treatment after age of 6 months? The most important is to start treatment early right? So if treatment is started after 6 months, treatment duration may be longer.

Line 63. Consider re-wording. There are no recent studies about incidence of CMT. The aim of this study was to examine incidence of CMT at our institution and associated factors.

Line 72. Consider replacing: We selected cases...we examined infants diagnosed with CMT....

Results: This section need to edited to read more smoothly. Please create paragraphs.

Line 94. Replace torticollis with CMT. Your first sentence in introduction says there are several types of torticollis so be precise.

Line 95. Incidence was 1.5% (NOT 1,5 %)

Line 98. Table 1. CI: Change to percentages.

Table 2.

Third column: It is labelled "congenital." Do you mean the most severe type? In introduction, you called it muscular with tumor. It is often called SCM nodule/tumor

Double check your numbers. In the results you state that n=118 (line 93), but in the table I am getting 121 for boys and girls, 117 for type of delivery, 115 for fetal presentation, 108 plagiocephaly etc. It all should add to 118.

Age at diagnosis: 0-1 month, 1-2 months, 2-3 months (remove>), 3 months and older.

101.  I m not familiar with the term eutocic and had to look it up. In USA the term used is SVD (spontaneous vaginal delivery). Dystoic delivery is also an unfamiliar term for me. It may be helpful to define these 2 terms for the reader. 

Discussion.

131. Expand this one sentence to make an introductory paragraph.

133. This is not a proper sentence.

138. I like how you link your findings with Cheng's study. Consider adding other citations to support your findings.

146. SCM muscle (add muscle at the end)

Continue to refine your discussion please.

Author Response

Thank you for affording us another opportunity to make our article read better for the global audience. We appreciate all the comments and suggestions that were made 

 I am excited by the aims of this article and all the work that was put into it. I believe that the topic is relevant and important to the field of pediatrics. However, the introduction, results and discussion are not well structured and need a lot of work to make it read and look professional. I recognize that English is not authors' first language, but the flow of the article itself needs major improvements.

Introduction: It reads a bit awkward and choppy.

First sentence: you say that it is the most common type of torticollis (what are the other types?).

Answer: We removed which is the most common so as not to generate confusion and added your suggestion.

You later discuss different types of CMT which may confuse the reader with the types mentioned in the first sentence.

Combine second and third sentence and improve the flow. It is a main point of your research so it would make it stronger to focus on the incidence first and maybe later talk about boys to girls ratio.

Answer: You are the 3rd reviewer and changes have already been made that have already been submitted, so I think your question is against the current version.

Was there is difference in how Stellwagen defined CMT vs other authors. Why is his incidence so high vs others? You can use it in your discussion as well.

Answer: It may be due to the type of diagnosis that was made in each study. It was added to the discussion.

4th sentence (line 38). Consider: Diagnosis of CMT is made via observation of alignment, cervical range of motion assessment, and palpation.

Answer: Done.

  1. This is a very confusing sentence: consider making it more streamlined and clear: For example: There are 3 types of CMT: postural, muscular and muscular with SCM mass.

Answer: This was another reviewer's suggestion, I can't change it.

Line 40. Consider changing word "rigidity" for "decreased flexibility)

Answer: Done.

Line 44. needs grammar edits. Consider: Is is greater strength or is the muscle overactive? Or both?

Answer: Children with CMT present a muscular imbalance (muscle overactive) in the neck muscles.13

This paragraph is only one sentence. Paragraph needs more than 2 sentences to be a paragraph. Consider combing with the section above.

Answer: Done.

Line 46. there are several theories not just one. Your first sentence is misleading. Rework this paragraph for better flow. Consider children who acquire CMT after birth, not in prenatal period. Your results show infants diagnosed later right?

Answer: Change made, but we found no theories for the etiology after birth.

Line 54-57. Needs to be edited for grammar and structure. Consider adding to your citations article by von Heideken at al.

von Heideken J, Green DW, Burke SW, Sindle K, Denneen J, Haglund-Akerlind Y, Widmann RF. The relationship between developmental dysplasia of the hip and congenital muscular torticollis. J Pediatr Orthop. 2006 Nov-Dec;26(6)

Answer: Done.

Lines 59-61. rewrite for grammar and clarity. It is confusing, especially 2nd sentence. Consider: is it about detection or initiation of treatment after age of 6 months? The most

important is to start treatment early right? So if treatment is started after 6 months, treatment duration may be longer.

Answer: “Early detection and treatment of CMT (before 1 month of age) have a high probability of recovering head posture symmetry (1.5 months). If CMT is detected from 6 months of age, the range of motion in the cervical region will gradually decrease, and a longer intervention period (between 9 and 10 months) may be necessary.1,8”

Line 63. Consider re-wording. There are no recent studies about incidence of CMT. The aim of this study was to examine incidence of CMT at our institution and associated factors.

Answer: “There are no recent studies about incidence of CMT. The aim of this study was to examine incidence of CMT at our institution, the type, the age of diagnosis and associated factors.”

I have to keep the type and age because it was a reviewer's suggestion.

Line 72. Consider replacing: We selected cases...we examined infants diagnosed with CMT....

Answer: It was not we, the authors, who examined it, it was the physiatrist.

Results: This section need to edited to read more smoothly. Please create paragraphs.

Answer: Done.

Line 94. Replace torticollis with CMT. Your first sentence in introduction says there are several types of torticollis so be precise.

Answer: Done.

Line 95. Incidence was 1.5% (NOT 1,5 %)

Answer: Done.

Line 98. Table 1. CI: Change to percentages.

Answer: We prefer to leave it that way, as the percentage is written in text before the table.

Table 2.

Third column: It is labelled "congenital." Do you mean the most severe type? In introduction, you called it muscular with tumor. It is often called SCM nodule/tumor

Answer: “as congenital torticollis with the presence of a nodule (torticollis with a tumor or a palpable fibrotic nodule in the sternocleidomastoid muscle).”

Double check your numbers. In the results you state that n=118 (line 93), but in the table I am getting 121 for boys and girls, 117 for type of delivery, 115 for fetal presentation, 108 plagiocephaly etc. It all should add to 118.

Answer: The gender values is correct (118).The dominant gender in CMT is male, with 77 (65.3%) and 41 (34.7%) female.”

The type of delivery value gives 117 because with 1 delivery mode could not be identified in the process - “Most deliveries of babies diagnosed with torticollis were eutocic (56; 47.5%), 43 (36.4%) cesarean section, 15 (12.7%) dystocic with the use of suction cup and 3 (2.5%) with the use of forceps, with 1 delivery mode could not be identified in the process.”

As well as other values do not give 118 because they did not have the reference in the process, as it is written in the texts before the tables.

Age at diagnosis: 0-1 month, 1-2 months, 2-3 months (remove>), 3 months and older.

Answer: Done.

  1. I m not familiar with the term eutocic and had to look it up. In USA the term used is SVD (spontaneous vaginal delivery). Dystoic delivery is also an unfamiliar term for me. It may be helpful to define these 2 terms for the reader.

Answer: We changed the eucocic term and added this information in the introduction section about dystocic delivery: Other risk factors involved in the development of congenital torticollis include breech presentation, multiple pregnancy and dystocic delivery (vaginal delivery using suction cups or forceps).4,6,15”

Discussion.

  1. Expand this one sentence to make an introductory paragraph.

Answer: This has already been changed with the other reviewers.

  1. This is not a proper sentence.

Answer: This has already been changed with the other reviewers.

  1. I like how you link your findings with Cheng's study. Consider adding other citations to support your findings.

Answer: There are not many studies that have verified the incidence of torticollis. There are studies that evaluated other variables related to torticollis, but do not present data on incidence.

  1. SCM muscle (add muscle at the end)

Answer: This abbreviation no longer exists.

Reviewer 3 Report

Dear authors,

thank you for this retrospective analysis of routine clinical data which confirms existing literature and strengthens the knowledge on the incidence of congenital torticollis. The presentation of your work needs some improvements, in particular with regard to the “associated risk factors” and the description of the data analysis. Causal attributions, associations, and interpretations must be clearly distinguished. Conclusions must fit the research question and should be cautious. You seem to have undeclared secondary aims, causal inferences on the pathology, and implications for treatment. The manuscript and its relevance would profit from integrating this line of argument from the beginning. 

Detailed review with suggestions:

Title

Consider bringing in the time of diagnosis and type of corticollis already in the title.

Abstract

L.12. Please specify what you mean by “associated risk factors”.

L 22 -24 The two statements in this long sentence would be clearer when stated in two sentences.

L 24 – 27 The conclusions open a new topic. Conclusions should refer to the presented study.

Keywords

Torticollis includes the muscular torticollis, omit. Maybe include “risk factor” and physiotherapy.

Introduction

L.31/32 “acquired in the prenatal period” This statement of the origin of CMT is apparently, from your later discussion and the cited literature, unclear (and difficult to verify).

L.34/35 here you cite a case report (not adequate)

L.36-37 It is normal and not surprising that epidemiological data are discrepant. It is adequate to point at differences between investigations, strengthen existing knowledge by adding an analysis of a large population sample, and discuss reasons for discrepancies.

L.54 I suggest a more cautious wording, CMT may be associated with one or more comorbidities……

L.63 You are not presenting a national study.

Methods

L.69 Please inform the reader about the intake region of the hospital or the percentage of newborns covered by the hospital’s birth register. Is it the only hospital in the region with a delivery ward? Are the hospital data representative of the local population or is there a separate private sector in Portugal? How frequent is delivery at home? Are newborns from other places presented in the clinic for examination? It is relevant to know how representative your data are.

L.72 Can you provide further information on the process of diagnosis? Is there a standard examination scheme for newborns in Portugal? Is the diagnosis made by a specialized medical doctor? How many doctors were involved in diagnosing CMT in the 5-year period? I want to raise your awareness that the reliability and validity of the diagnosis have a major influence on your data, and may explain potential differences with other studies. Therefore, provide the information that you have on the local process of diagnosing CMT.

L.81-90 The report of the data analysis is incomplete and not professional. First report which variables were extracted from the newborns’ medical history. Here, the “potential risk factors” must be clarified. Then the descriptive statistics, preferably using reporting standards. It would be valuable to provide the definition of the outcome measures, not every reader is an epidemiologist. Does the term “qualitative variable” refer to the scaling? The description of the statistical procedures should allow for the replication of the study.

Results

Table 1: please provide also the numbers of births and diagnosed cases per year and over the 5 years. Use the same number of decimals consistently.

I would prefer the result statements in a single paragraph, ordered content-wise; without a break after each statement.

L.108-110 Please, for clarity use exactly the diagnostic categories that you explained in the introduction.

Table 2 better use n.a. (not applicable) than an asterisk (usually an indication of significance). What was tested?

Discussion

With an incidence of 1.5, why do you state “a very common pathology”? Why do you refer firstly to the etiology of CMT? This was not the scope of your study. At the start of the discussion, you should summarize your major findings. In the following paragraphs, each finding is discussed in comparison with the existing literature. This way, you structure the discussion more clearly.

L. 208… This final paragraph of the discussion appears to include your interpretation of a relationship between the time of diagnosis and the category of CMT. State it accordingly and be cautious to distinguish interpretations from evidence-based facts.

Conclusions are too long. You should not repeat your findings in detail.

Are physiotherapists involved in diagnosing newborns? Why should only physiotherapists plan a prevention strategy? Isn’t this an interprofessional matter?

The aspects you are referring to in the final two paragraphs of the conclusions open a new theme that has not been addressed before. The current final paragraph is not good for closing your report. You should integrate this line of argument already in the introduction and the aim of the study.

Author Response

Thank you for affording us another opportunity to make our article read better for the global audience. We appreciate all the comments and suggestions that were made by two reviewers. I hope we have made the revisions to your satisfaction.

REVISOR 2

Dear authors,

thank you for this retrospective analysis of routine clinical data which confirms existing literature and strengthens the knowledge on the incidence of congenital torticollis. The presentation of your work needs some improvements, in particular with regard to the “associated risk factors” and the description of the data analysis. Causal attributions, associations, and interpretations must be clearly distinguished. Conclusions must fit the research question and should be cautious. You seem to have undeclared secondary aims, causal inferences on the pathology, and implications for treatment. The manuscript and its relevance would profit from integrating this line of argument from the beginning. 

Detailed review with suggestions:

Title

Consider bringing in the time of diagnosis and type of corticollis already in the title.

Answer: Incidence of congenital muscular torticollis in babies from southern Portugal: types, age of diagnosis and risk factors

Abstract

L.12. Please specify what you mean by “associated risk factors”.

Answer: Withdrawn.

L 22 -24 The two statements in this long sentence would be clearer when stated in two sentences.

Answer: Modified.

L 24 – 27 The conclusions open a new topic. Conclusions should refer to the presented study.

Answer: Withdrawn.

Keywords

Torticollis includes the muscular torticollis, omit. Maybe include “risk factor” and physiotherapy.

Answer: We prefer not to put physical therapy.

Keywords: congenital muscular torticollis, babies, incidence, epidemiology, risk factor

Introduction

L.31/32 “acquired in the prenatal period” This statement of the origin of CMT is apparently, from your later discussion and the cited literature, unclear (and difficult to verify).

Answer: Withdraw.

L.34/35 here you cite a case report (not adequate)

Answer: The reference was only used to define the concept of CMT.

L.36-37 It is normal and not surprising that epidemiological data are discrepant. It is adequate to point at differences between investigations, strengthen existing knowledge by adding an analysis of a large population sample, and discuss reasons for discrepancies.

Answer: We found few studies that evaluated the incidence, making the discussion poorer.

L.54 I suggest a more cautious wording, CMT may be associated with one or more comorbidities……

Answer: Changed.

L.63 You are not presenting a national study.

 Answer: No national studies on the incidence were found.

Methods

L.69 Please inform the reader about the intake region of the hospital or the percentage of newborns covered by the hospital’s birth register. Is it the only hospital in the region with a delivery ward? Are the hospital data representative of the local population or is there a separate private sector in Portugal? How frequent is delivery at home? Are newborns from other places presented in the clinic for examination? It is relevant to know how representative your data are.

Answer: We added these paragraphs:

The Algarve region occupies an area of 4,996 km2 and is home to 467,495 inhabitants (2021), concentrating 4.5% of the resident population in Portugal, which comprises only one subregion, consisting of 16 municipalities and divided into 67 parishes.

Under the public health system, there are 2 university hospitals (CHUA) with delivery and maternity rooms: one located in the city of Portimão and the other in Faro. On a private level, there is only one hospital in Faro.

We also added this: “The study has other limitation, as the fact that some babies were not referred for consultation in the Algarve University Hospital Center, Portimão unit, having been diagnosed in another hospital or clinic, thus not being counted in the study. Another limitation is the fact that some babies were born in this hospital because the mother was here at the time of delivery or some parents may have changed residence and gone to another place for the pediatric consultation.”

L.72 Can you provide further information on the process of diagnosis? Is there a standard examination scheme for newborns in Portugal? Is the diagnosis made by a specialized medical doctor? How many doctors were involved in diagnosing CMT in the 5-year period? I want to raise your awareness that the reliability and validity of the diagnosis have a major influence on your data, and may explain potential differences with other studies. Therefore, provide the information that you have on the local process of diagnosing CMT.

Answer: There is no standard of examination for torticollis. The diagnosis is made by a physiatrist. Over the 5 years it was not always the same doctor who made the diagnosis.

L.81-90 The report of the data analysis is incomplete and not professional. First report which variables were extracted from the newborns’ medical history. Here, the “potential risk factors” must be clarified. Then the descriptive statistics, preferably using reporting standards. It would be valuable to provide the definition of the outcome measures, not every reader is an epidemiologist. Does the term “qualitative variable” refer to the scaling? The description of the statistical procedures should allow for the replication of the study.

 Answer: I apologize, but I disagree with your opinion.

I will not be describing all the variables analyzed here, as they will appear in the results.

Student risk factors are described in table 2. I will not mention them here, it is unnecessary.

What would reporting standards be?

What definitions would be outcome measures?

Qualitative variables are variables that are not numerical and that cannot be analyzed with the chi-square test.

Only those who have statistical knowledge will replicate the study and the information given is enough to do so.

Results

Table 1: please provide also the numbers of births and diagnosed cases per year and over the 5 years. Use the same number of decimals consistently.

Answer: Done.

I would prefer the result statements in a single paragraph, ordered content-wise; without a break after each statement.

Answer: Done.

L.108-110 Please, for clarity use exactly the diagnostic categories that you explained in the introduction.

Answer: Done.

Table 2 better use n.a. (not applicable) than an asterisk (usually an indication of significance). What was tested?

Answer: n.a. does not apply. The * indicates statistical significance if it was a p-value value. The * can also refer to some observation, which is what we did.

 Discussion

With an incidence of 1.5, why do you state “a very common pathology”? Why do you refer firstly to the etiology of CMT? This was not the scope of your study. At the start of the discussion, you should summarize your major findings. In the following paragraphs, each finding is discussed in comparison with the existing literature. This way, you structure the discussion more clearly.

Answer: Done. It was a mistake.

  1. 208… This final paragraph of the discussion appears to include your interpretation of a relationship between the time of diagnosis and the category of CMT. State it accordingly and be cautious to distinguish interpretations from evidence-based facts.

Answer: We agree. Withdraw.

Conclusions are too long. You should not repeat your findings in detail.

Answer: Changed.

Are physiotherapists involved in diagnosing newborns? Why should only physiotherapists plan a prevention strategy? Isn’t this an interprofessional matter?

Answer: No. We changed the abstract: “Despite presenting a low incidence, it is important to mention the importance of the health profession intervention, as a physiotherapist, in the implementation of prevention strategies.”

The aspects you are referring to in the final two paragraphs of the conclusions open a new theme that has not been addressed before. The current final paragraph is not good for closing your report. You should integrate this line of argument already in the introduction and the aim of the study.

Answer: Withdraw.

Round 2

Reviewer 1 Report

The edits that were made improved the manuscript.  Although the paragraph in the conclusion regarding physiotherapist's responsibilities for prevention  was deleted, a sentence remains in the abstract indicating the importance of the physiotherapist in prevention CMT - inconsistent?  

The issue that makes this manuscript less valuable to readers, in my opinion, is the retrospective nature of the data, the small percentage of the population studied and the lack of control over the true incidence in the overall population.  

Although the authors responded to the question about positioning practices in the answers to the reviewers, it would be helpful to understanding the findings that the most cases were postural if those positioning practices were discussed in the article.  

Author Response

The edits that were made improved the manuscript.  Although the paragraph in the conclusion regarding physiotherapist's responsibilities for prevention  was deleted, a sentence remains in the abstract indicating the importance of the physiotherapist in prevention CMT - inconsistent? 

Answer: Withdrawn.

The issue that makes this manuscript less valuable to readers, in my opinion, is the retrospective nature of the data, the small percentage of the population studied and the lack of control over the true incidence in the overall population. 

Although the authors responded to the question about positioning practices in the answers to the reviewers, it would be helpful to understanding the findings that the most cases were postural if those positioning practices were discussed in the article.

Answer: Thank you for your comment, but I think it makes no sense to be discussing these practices, as it was not the purpose of the study. If we were studying the effects of positioning in improving torticollis, or as prevention, it would make sense, but that was not the objective, and I think it would be off topic.

Reviewer 2 Report

I think this paper still need significant improvement before it is publishable. 

Please include APTA CMT CPG 2018 in your references. I think it will help you with your discussion and conclusions. 

Intro:

Line 36. pediatric musculoskeletal conditions seen in infancy, defined as (remove being)

39-41. I like the change. Please combine sentences to create a paragraph.

Line 44. There are 3 types of CMT: torticollis with....

Line 49. Children with CMT present with a cervical muscle strength imbalance 

Line 51. but it is still not fully known (remove unknown)

Line 54. remove "however"

Line 60-68. needs to be rewritten, does not read well.

Line 72. The aim of this study was to examine incidence of CMT at our institution, type distribution, infant's age at a diagnosis, and associated factors.

Discussion: 

It looks better, but still needs to be streamlined. Make your conclusions stronger by combining several articles and summarizing their results.

176-184 Should be one paragraph

I like your paragraph 185-190:)

Line 204. I like your point. I think clinician's skill at palpating the SCM may play a role, or is palpation always included in the examination?

Line 207. Change always to most often/predominantly

Line 208. when they have poor head control

216-226. Combine into one paragraph

234-246 . Create sections called Study Limitations.

Line 241. Edit to read> some infants were......, and have been diagnosed at another clinic

Line 243-246. Confusing....shorten and clarify please

Conclusions.

This section should summarize your findings not introduce new ideas.

Line 250. the most prevalent being postural CMT

Line 256-260. It is a new idea not introduced on the discussion. I like it but I think you need to develop it further (use APTA CMT CPG as a guide).

Author Response

Thanks for your corrections and suggestions.

I think this paper still need significant improvement before it is publishable. 

Please include APTA CMT CPG 2018 in your references. I think it will help you with your discussion and conclusions. 

Answer: We added this reference.

Intro:

Line 36. pediatric musculoskeletal conditions seen in infancy, defined as (remove being)

Answer: Done.

39-41. I like the change. Please combine sentences to create a paragraph.

Answer: Done.

Line 44. There are 3 types of CMT: torticollis with....

Answer: Done.

Line 49. Children with CMT present with a cervical muscle strength imbalance 

Answer: Done.

Line 51. but it is still not fully known (remove unknown)

Answer: Done.

Line 54. remove "however"

Answer: Done.

Line 60-68. needs to be rewritten, does not read well.

Answer: Done.

Line 72. The aim of this study was to examine incidence of CMT at our institution, type distribution, infant's age at a diagnosis, and associated factors.

 Answer: Done.

Discussion: 

It looks better, but still needs to be streamlined. Make your conclusions stronger by combining several articles and summarizing their results.

176-184 Should be one paragraph

 Answer: Done.

I like your paragraph 185-190:)

Line 204. I like your point. I think clinician's skill at palpating the SCM may play a role, or is palpation always included in the examination?

Answer: We added thois: “For example, the clinician's ability to palpate the SCM can also be crucial to a good diagnosis.”

Line 207. Change always to most often/predominantly

 Answer: Done.

Line 208. when they have poor head control

 Answer: Done.

216-226. Combine into one paragraph

 Answer: Done.

234-246 . Create sections called Study Limitations.

 Answer: We did not create a section for limitations in the study, as it is not required in the rules of this journal, but I put the limitations together in a single paragraph.

Line 241. Edit to read> some infants were......, and have been diagnosed at another clinic

 Answer: Done.

Line 243-246. Confusing....shorten and clarify please

Answer: “Another limitation is the fact that some babies may have been born in this hospital, but not have gone to the pediatric consultation in this hospital. Examples: at the time of delivery this was the closest hospital, but the parents did not live here; some parents may have changed residence and gone to another location for the pediatric consultation; and some parents may have preferred to go to the consultation at a clinic or hospital particular.”

Conclusions.

This section should summarize your findings not introduce new ideas.

Answer: We made some changes.

 It looks better, but still needs to be streamlined. Make your conclusions stronger by combining several articles and summarizing their results.

Answer: We made some changes.

Line 250. the most prevalent being postural CMT

 Answer: Done.

Line 256-260. It is a new idea not introduced on the discussion. I like it but I think you need to develop it further (use APTA CMT CPG as a guide).

Answer: We made some changes and added this reference.

Reviewer 3 Report

Dear authors,

Thank you for the rapid resubmission of this work. The references to physiotherapy are now integrated in an appropriate manner. Major points in the first part of the work have been revised well, thank you.

With regard to the statistical report, the current version is scientifically meager. If you talk about “data”, you should specify which data are meant, which are the main outcomes, and which associated factors. If you argue that all collected data are identifiable in table 2, you should state this in the methods. I am certain that you examined for normal distribution before you decided on the appropriate statistical test. Therefore, you should first report the Kolmogorov-Smirnow test. Which data were skewed? All? It is appropriate to state the results of the KS test already here to support the choice of the inferential tests. Then your inferential statistics. Which data were examined using the Kruskal Wallis test and which using the Chi-square test? [Any statistics book or google will assist you in proper statistical reporting, and not much change is required for providing more succinct information, e.g. https://statistics.laerd.com/spss-tutorials/kruskal-wallis-h-test-using-spss-statistics.php  for the KW-test. ]

You’ve stated in your response that my suggestions regarding table 1 have been addressed, but the submitted table is exactly the same as in the first version.

L138 / 139: the aspect of laterality of plagiocephaly would be more informative when combined with the laterality of the torticollis. The current statement is correct, my suggestion would improve the information from the descriptive results.

Discussion

Structuring the discussion with a space between the paragraphs would improve the structure and readability.

L 157: you interpret an incidence of 1.5% as high. According to which interpretation scheme (reference)?

L 168 – 174 contains a single sentence consisting of at least five statements. Please, rephrase.

L 179 Didn’t you state before that there are no national studies?

L188 – 196, the reliability of the diagnosis of the type of torticollis may play a major role in the differences between studies. How reliable is the diagnosis? If the type of torticollis is the most inconsistent finding between different studies (and sex/ laterality is more consistent), this would suggest that the diagnosis of the type of torticollis is not highly reliable.

L214-215 Do boys have a larger head? Men have [1], but is this true also for babies?

L231-239 The last sentence of this paragraph should start the paragraph to avoid the impression that these causal inferences are evidence-based.

1.            Clemens, H.J. Das Kopfgewicht des Menschen - ein biomechanisches Problem. Arch. Orthop. Unfallchir. 1972, 73, 220-228.

Author Response

Thank you for the rapid resubmission of this work. The references to physiotherapy are now integrated in an appropriate manner. Major points in the first part of the work have been revised well, thank you.

With regard to the statistical report, the current version is scientifically meager. If you talk about “data”, you should specify which data are meant, which are the main outcomes, and which associated factors. If you argue that all collected data are identifiable in table 2, you should state this in the methods. I am certain that you examined for normal distribution before you decided on the appropriate statistical test. Therefore, you should first report the Kolmogorov-Smirnow test. Which data were skewed? All? It is appropriate to state the results of the KS test already here to support the choice of the inferential tests. Then your inferential statistics. Which data were examined using the Kruskal Wallis test and which using the Chi-square test? [Any statistics book or google will assist you in proper statistical reporting, and not much change is required for providing more succinct information, e.g. https://statistics.laerd.com/spss-tutorials/kruskal-wallis-h-test-using-spss-statistics.php  for the KW-test. ]

Answer: I'm very sorry, but we only use chi-square as a statistical inference. I'm writing another article at the same time and I got confused. I stated that the variables that were worked on in the Chi-square are shown in table 2.

You’ve stated in your response that my suggestions regarding table 1 have been addressed, but the submitted table is exactly the same as in the first version.

Answer: Weird. This was the changed table submitted in the new version.

   Table 1: Incidence proportion by years and cumulative incidence

Year of data collection

2016

2017

2018

2019

2020

5-years period

Numbers of births

1,283

1,302

1,359

1,363

1,258

6,565

Diagnosed cases

25

17

22

31

23

118

Incidence proportion

0.019

CI:  0.013-0.029

0.013

CI: 0.008-0.021

0.027

CI: 0.019-0.037

0.023

CI: 0.016-0.032

0.018

CI: 0.012-0.027

0.015

CI: 0.013-0.018

L138 / 139: the aspect of laterality of plagiocephaly would be more informative when combined with the laterality of the torticollis. The current statement is correct, my suggestion would improve the information from the descriptive results.

Answer: We added this table in the results sections and changed the value in discussion section.

Plagiocephaly

Laterality of torticollis

Total

Right

Left

Absent

34 (59.6%)

29 (47.5%)

63 (53.4%)

Present

19 (33.3%)

29 (47.5%)

48 (40.7%)

Without this reference

4 (7%)

3 (4.9%)

7 (5.9%)

Regarding the laterality of plagiocephaly obtained, in this study, the right side is the most predominant (57%), confirming the prevalence of torticollis on the left ((47.5%).

Discussion

Structuring the discussion with a space between the paragraphs would improve the structure and readability.

Answer: Done.

L 157: you interpret an incidence of 1.5% as high. According to which interpretation scheme (reference)?

Answer: You must not have received the new version. “The data presented revealed a low incidence of CMT”

L 168 – 174 contains a single sentence consisting of at least five statements. Please, rephrase.

Answer: I put it in another paragraph.

L 179 Didn’t you state before that there are no national studies?

Answer: There are no national studies on the incidence of torticollis, but there are national studies with other data.

L188 – 196, the reliability of the diagnosis of the type of torticollis may play a major role in the differences between studies. How reliable is the diagnosis? If the type of torticollis is the most inconsistent finding between different studies (and sex/ laterality is more consistent), this would suggest that the diagnosis of the type of torticollis is not highly reliable.

Answer: We added this phrase: “The reliability of the diagnosis of the type of torticollis may play a major role in the differences between studies.”

L214-215 Do boys have a larger head? Men have [1], but is this true also for babies?

Answer: “suggesting that boys have larger heads and less flexible than girls”.

L231-239 The last sentence of this paragraph should start the paragraph to avoid the impression that these causal inferences are evidence-based.

 Answer: Done.
